# Generic wound signals initiate regeneration in missing-tissue contexts

Suthira Owlarn [1,2], Felix Klenner [3], David Schmidt [1,2], Franziska Rabert[1,2], Antonio Tomasso [1,2], Hanna Reuter[1,2], Medhanie A. Mulaw[4], Sören Moritz[1], Luca Gentile[1,5], Gilbert Weidinger [3] & Kerstin Bartscherer [1,2]

Despite the identification of numerous regulators of regeneration in different animal models, a fundamental question remains: why do some wounds trigger the full regeneration of lost body parts, whereas others resolve by mere healing? By selectively inhibiting regeneration initiation, but not the formation of a wound epidermis, here we create headless planarians and finless zebrafish. Strikingly, in both missing-tissue contexts, injuries that normally do not trigger regeneration activate complete restoration of heads and fin rays. Our results demonstrate that generic wound signals have regeneration-inducing power. However, they are interpreted as regeneration triggers only in a permissive tissue context: when body parts are missing, or when tissue-resident polarity signals, such as Wnt activity in planarians, are modified. Hence, the ability to decode generic wound-induced signals as regeneration-initiating cues may be the crucial difference that distinguishes animals that regenerate from those that cannot.

[1] Max Planck Institute for Molecular Biomedicine, Von-Esmarch-Str. 54, 48149 Münster, Germany. [2] Medical Faculty, University of Münster, Albert-Schweitzer-Campus 1, 48149 Münster, Germany. [3] Institute of Biochemistry and Molecular Biology, Ulm University, Albert-Einstein-Allee 11, 89081 Ulm, Germany. [4] Institute of Experimental Cancer Research, Ulm University, Albert-Einstein-Allee 11, 89081 Ulm, Germany. [5]Present address: Pluripotency and Regeneration Group, Fraunhofer Institute for Biomedical Engineering, Ensheimerstr. 48, 66386 St. Ingbert, Germany

Complete restoration of lost body parts is the ultimate goal of regenerative medicine. In humans, restorative responses following dramatic injuries are limited, generally leading to wound healing with scarring rather than regeneration of the missing body parts. Model organisms capable of regenerating are therefore of utmost importance for understanding the mechanisms of successful regeneration programs. In such animals, injuries that remove body parts and require regeneration (R-wounds) first trigger formation of a wound epithelium, and subsequently lead to the activation of key regenerative processes, such as blastema formation and the reconstruction of patterned tissues[1–4]. In contrast, injuries that require only healing (H-wounds) and do not involve significant tissue loss do not initiate the regenerative response after formation of a wound epithelium[3,5].

A plethora of global gene expression analyses has revealed dramatic changes in transcription following injury and enabled the identification and classification of hundreds of genes that are activated during the first few hours after tissue amputation in several regeneration model systems[4,6–9]. In addition, both unbiased approaches, like genetic screens, and targeted functional studies have identified many signaling pathways and other molecular players involved in the regulation of regenerative growth and patterning[10–12]. However, despite efforts such as comparative transcriptomics between H-wounds and R-wounds to identify amputation-specific genes[8,9], little is known about the molecular events that distinguish wound responses that trigger regeneration from those that trigger wound healing.

Identifying regeneration initiation signals remains a key challenge in regeneration research. Notably, the salamander accessory limb model has confirmed the critical role of nerves in initiating blastema formation in urodeles, and the role of positional cues in patterning and growing a blastema into a limb[13]. However, fundamental questions about the nature of regeneration initiation signals have remained unanswered. In particular, it is unclear whether regeneration initiation signals are induced exclusively after R-wounds (Fig. 1, Hypothesis I) or generically after all types of injuries, including H-wounds (Fig. 1, Hypothesis II). In the first model, regeneration occurs solely after R-wounds because they are the only wound type in which regeneration initiation signals are induced. In the latter model, different tissue contexts (generated by either H- or R-wounds) interpret the same generic wound signals differently, leading to either regeneration at R-wounds or mere wound healing at H-wounds (Fig. 1). Answering this fundamental question has important implications for regenerative medicine: depending on which hypothesis is more accurate, providing damaged tissues with initiation signals may or may not be sufficient for promoting regenerative processes. However, a major limitation in testing these models has been the inability to uncouple the process of wounding from that of tissue loss.

Here we overcome this limitation by temporally separating regeneration initiation from tissue loss in the highly regenerative planarian and zebrafish models. In planarians, pharmacological inhibition of ERK activation severely interfered with regenerative responses as early as 3 h post-amputation, establishing ERK activation as the earliest signal essential for regeneration initiation in planarians. Similarly, transient inhibition of FGF or Wnt signaling in zebrafish permanently prevented regeneration of amputated fin rays, resulting in tissues that heal wounds but fail to regenerate. Strikingly, infliction of simple H-wounds, which normally do not induce regeneration, in these missing-tissue contexts triggered full regeneration of all missing structures, including the central nervous system in planarians and bones in fish. Additionally, H-wounds inflicted on otherwise intact planarians after manipulating Wnt signaling activity also induced organized outgrowths, such as ectopic heads, suggesting that

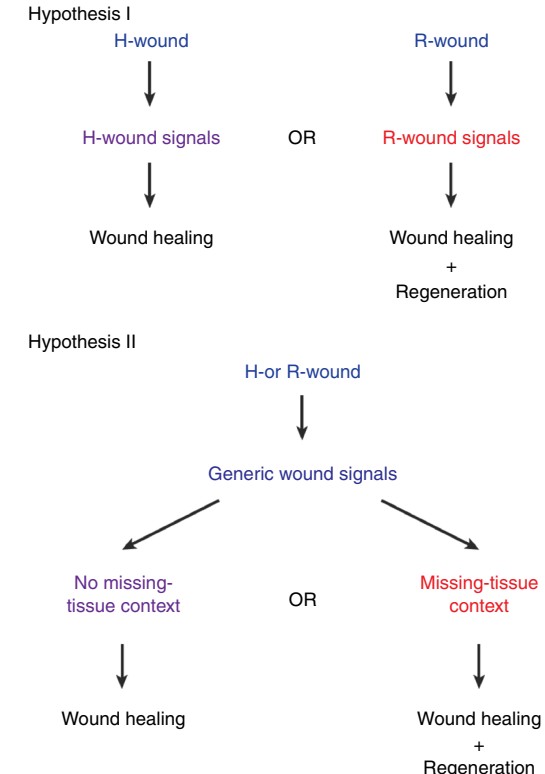

**Fig. 1** Two hypotheses for generating regeneration initiation signals. Hypothesis I: regeneration-initiation signals are induced exclusively after R-wounds, such as amputations; H-wounds, such as incisions or other epidermal wounds, activate different signals leading to wound healing but not to a subsequent regenerative response. Hypothesis II: the same signals are induced generically after all types of injuries; the tissue contexts in which the signals are induced determine whether regeneration occurs

Wnt-mediated positional cues play an important role in regulating the missing-tissue context. Together, these results demonstrate that generic signals activated by different types of wounds are sufficient for initiating regeneration.

## Results

**Transient ERK inhibition permanently blocks regeneration.** Planarians of the species *Schmidtea mediterranea* are a powerful model organism for studying regeneration, as they are capable of regenerating any part of their body, including a functional brain, within a week after amputation[14,15]. Following tissue amputation, planarian adult stem cells (also referred to as neoblasts) migrate to the wound site, proliferate and differentiate to replace lost cell types. These processes, as well as stem cell-dependent formation of regeneration poles and tissue remodeling (reviewed in ref. [12,16,17]), are induced after R-wounds, but not after H-wounds[3]. To investigate whether regeneration initiation signals are activated generically at all wounds (Hypothesis II in Fig. 1) or exclusively at R-wounds (Hypothesis I in Fig. 1), we set out to establish an assay in which the regeneration-inducing capacities of newly inflicted H- and R-wounds could be tested.

As extracellular regulated kinase (ERK) signaling regulates early processes during regeneration in several animals[18–20], we modulated this pathway as a candidate approach for blocking regeneration initiation, while allowing wound healing, after tissue amputation. ERK signaling has been suggested to regulate stem cell differentiation in planarians[21], a process that plays an important role particularly during later phases of regeneration

when new tissues are formed. However, we observed a dramatic increase in activated ERK (pERK) already within minutes after amputation (Fig. 2a). pERK levels returned to baseline within 24 h, suggesting that ERK functions mainly in the early stages of planarian regeneration, independent of differentiation processes. Interestingly, we detected similar levels of wound-induced pERK levels in planarians depleted of stem cells by γ-irradiation, and after inhibition of protein synthesis by cycloheximide treatment (Fig. 2b). These results suggest that ERK is rapidly activated in a stem cell-independent manner by injury signals that are not newly synthesized proteins.

In order to have strict temporal control over ERK activity, we employed PD0325901 (PD) and U0126 (U0), two potent but structurally independent inhibitors of the ERK-activating kinase (MEK)[22,23] (Fig. 2c, d, Supplementary Fig. 1a, b). Treatment of planarians with PD or U0 robustly inhibited regeneration at anterior and posterior amputation sites, including formation of the anterior and posterior regeneration poles[24,25] and

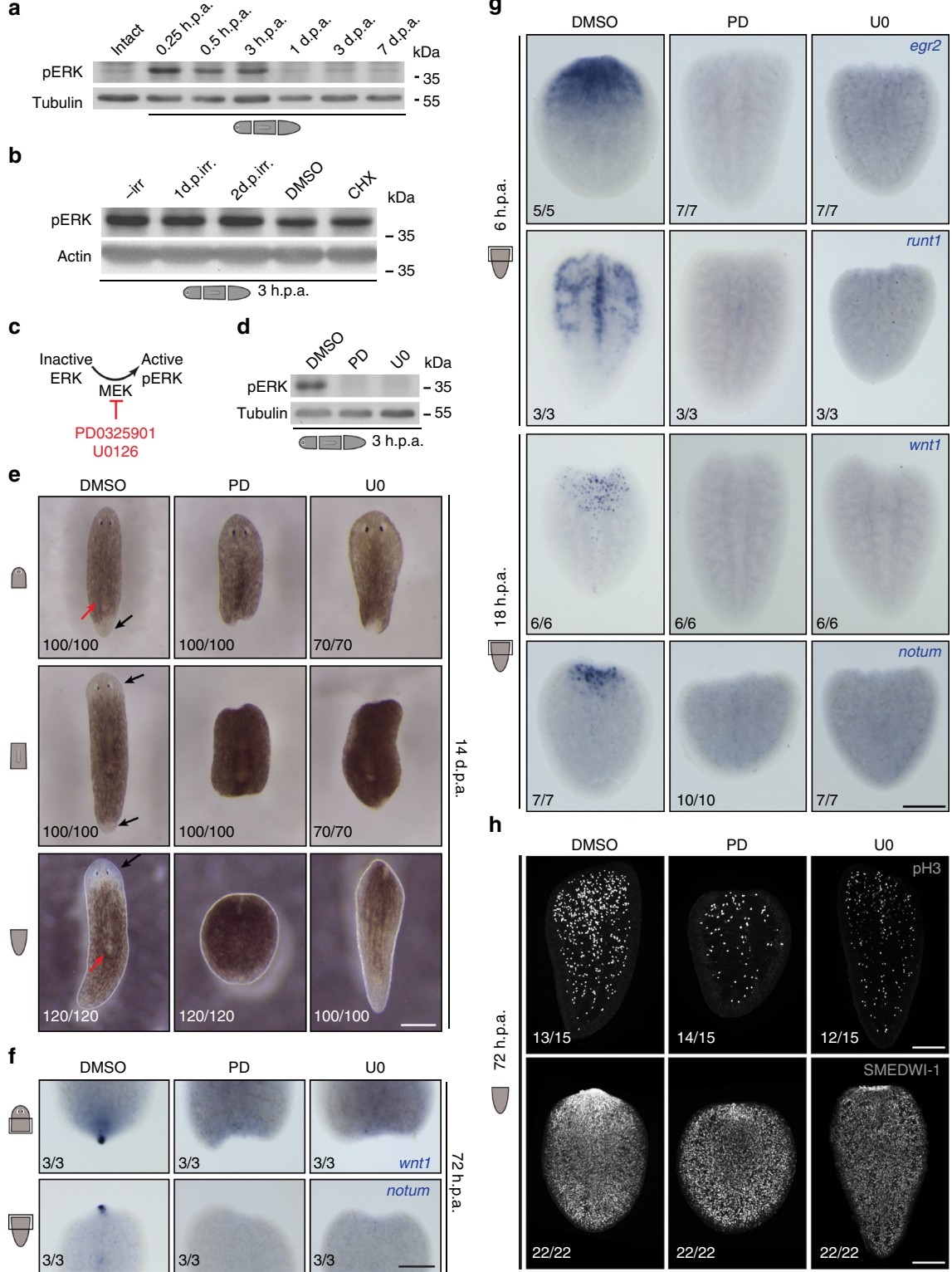

differentiation of new tissues (Fig. 2e, f, Supplementary Fig. 1c). Notably, ERK signaling was required within the first 12 h of regeneration (Supplementary Fig. 2), and early regeneration markers, such as *egr2*, *runt*, *wnt1*, and *notum*[6,7,24–26], were not induced after PD or U0 treatment (Fig. 2g, Supplementary Fig. 3a, b). Moreover, the regeneration-associated boost in proliferation[3], accumulation of multipotent cells at the amputation site[3], and remodeling of existing tissues[5] were all strongly reduced in drug-treated animals (Fig. 2h, Supplementary Figs. 1c, 3c–f). Among the pERK-dependent wound-induced genes was *follistatin (fst)* (Supplementary Fig. 4a), which promotes regeneration initiation by inhibiting Activin1 and 2[26]. We found that ERK inhibition increased the penetrance of the *fst* RNAi phenotype (Supplementary Fig. 4b). Moreover, *activin1/2* RNAi led to a partial rescue of regeneration in PD/U0 treated planarians. Together our results indicate that ERK activation by a yet unknown factor is the most upstream step of regeneration initiation identified in planarians so far, leading to *fst* activation and subsequent inhibition of Activin signaling.

Interestingly, ERK was similarly activated in response to amputations (R-wounds) and to incisions that heal, but do not trigger regenerative responses[3] (H-wounds) (Supplementary Fig. 5a). Yet, ERK inhibition did not prevent the immediate wound response involving wound closure and restoration of the epidermis (Supplementary Fig. 5b–c), activation of the stress-response genes *egrl1*, *mex3*, and *traf2*[4] (Supplementary Fig. 5d–f), and generic wound-induced apoptosis[3,5] (Supplementary Fig. 5g). Notably, like DMSO control planarians, drug-treated animals showed a significant increase in generic wound-induced proliferation (Supplementary Fig. 3d), which occurs within a few hours in response to both H- and R-wounds. This is in contrast to the regeneration-specific second wave of proliferation, which was strongly dependent on ERK activation (Fig. 2h, Supplementary Fig. 3c, d). Proliferation in uninjured planarians was slightly diminished after 7 days of extended treatment with either PD or U0 (Supplementary Fig. 5h), suggesting a minor role of ERK in homeostatic cell division. In summary, our results show that ERK activation is essential for regeneration initiation, but is dispensable for wound healing in planarians (Fig. 3a).

**Generic wound signals can initiate head regeneration.** As previous studies have shown that PD is more selective than U0 in other systems[27], we continued with further investigations using PD only. Tail fragments treated with PD for 5 days showed no signs of regeneration even 18 days after inhibitor removal (Fig. 3b), demonstrating that transient inhibition of the ERK pathway during the initiation stage was sufficient for permanently blocking regeneration. As ERK activation is required for initiating regeneration but not wound healing (Fig. 2, Supplementary Figs. 3–5), and the effect of PD on pERK levels is reversible (Supplementary Fig. 1b), we then asked whether PD-treated non-regenerating tissue fragments could maintain the intrinsic ability to regenerate after drug removal. Indeed, when new initiation

signals were induced in PD-treated tail fragments by re-amputation, either immediately or 5 days after drug removal, the animals formed blastemas and subsequently eyes (Fig. 3b). This indicates that although these tail fragments only healed, they had preserved the dormant competence to regenerate. These "dormant" fragments therefore constitute a powerful tool for temporally separating loss of body parts from the infliction of injuries, enabling the evaluation of the regeneration-inducing capacity of different types of wounds.

When intact planarians are injured, R-wounds initiate a regenerative response, whereas H-wounds heal by epidermal closure (Fig. 3a). However, it is unclear whether these types of wounds induce different signals, or whether the presence or absence of tissue leads to a differential interpretation of the same generic wound-induced signals (Fig. 1). It was recently shown that a common transcriptional response is initially activated after R- and H-wounds in planarians[4], raising the possibility that a generic wound response might be required for regeneration. We therefore exploited dormant tails as a unique missing-tissue model and asked whether an H-wound would be sufficient for inducing regeneration. If regeneration-initiation signals were specific to R-wounds, then infliction of H-wounds would not trigger regeneration in dormant tails (Fig. 1, Hypothesis I). Alternatively, if both R- and H-wounds were to induce regeneration-initiation signals, then even an H-wound would activate the full regenerative response (Fig. 1, Hypothesis II).

Remarkably, unlike uninjured controls, dormant tails injured by a simple H-wound induced blastema formation and other key regenerative events typically observed only after R-wounds. These events were ERK-dependent and included increased proliferation and accumulation of stem cells at the wound, tissue remodeling, expression of anterior markers, and restoration of the nervous and digestive systems (Fig. 3c–f, Supplementary Fig. 6). These animals also recovered functionally, as shown by resumption of feeding and locomotive behaviors (Fig. 3g). The effect was amplified when several H-wounds were inflicted at the same time (Fig. 3c–g, Supplementary Fig. 6). Notably, ERK signaling was activated in all re-wounding scenarios in dormant tails in absence of PD (Fig. 3h), and the same wounds failed to initiate regeneration under continued PD treatment (Fig. 3c–g, Supplementary Fig. 6). Strikingly, even 58 days after drug removal, H-wounding of dormant tails effectively rescued regeneration (Supplementary Fig. 7). Taken together, these results show that H-wounds can elicit the same regenerative responses as R-wounds, but only in a missing-tissue context.

**Generic wounds induce outgrowths in tissues with altered polarity.** To confirm the regeneration-inducing capacity of H-wounds, we searched for independent tissue paradigms in which H-wounds could activate regenerative responses. In planarians, the Wnt/β-catenin pathway is required for regeneration polarity and to maintain anterior-posterior polarity during homeostasis. It is therefore an excellent candidate pathway for modulating

**Fig. 2** Transient inhibition of ERK activation blocks regeneration initiation in planarians. **a** Phosphorylated ERK levels dramatically increased within 15 min post-amputation. **b** Amputation-induced ERK activation occurred even after depletion of stem cells by γ-irradiation (1 or 2 days post-irradiation (d.p.irr)) and after inhibition of protein synthesis with cycloheximide (CHX). **c** MEK activates ERK by phosphorylation (pERK); MEK inhibitors PD0325901 (PD) and U0126 (U0) prevent ERK activation by inhibiting MEK. **d** Treatment with PD and U0 effectively reduced pERK levels. **e**, **f** Inhibition of ERK activity severely (PD heads 44/100, trunks 33/100, tails 3/120; U0 heads 43/70, trunks 47/70, tails 11/70) or completely (PD heads 56/100, trunks 67/100, tails 117/120; U0 heads 27/70, trunks 23/70, tails 59/70) inhibited formation of blastemas (**e**) and anterior (*notum* expression in lower panel) and posterior (*wnt1* expression in upper panel) regeneration poles (**f**). Black arrows: regeneration of new tissues in the blastema; red arrows: remodeling of existing tissues. **g**, **h** Other key regenerative responses, such as expression of early wound-induced genes (**g**), as well as stem cell proliferation (pH3, phospho-Histone H3 as a marker for mitotic cells) and accumulation of stem cells and progeny (SMEDWI-1) (**h**) were also severely affected by ERK inhibition. Sample numbers indicated in each image. h.p.a., hours post-amputation; d.p.a., days post-amputation; DMSO was used as the solvent control; scale bars: 200 μm

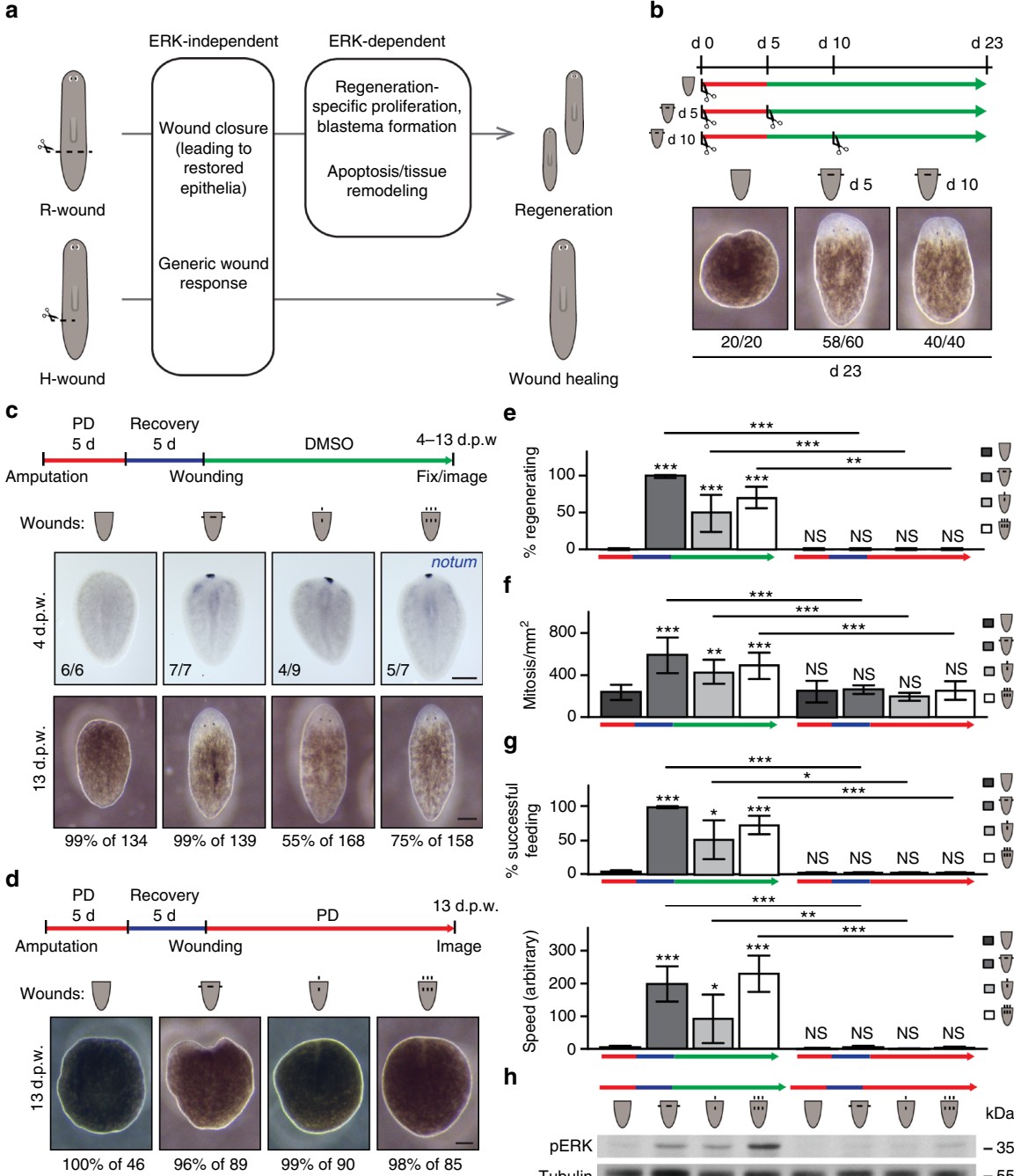

**Fig. 3** H-wounds inflicted on dormant tails induce regeneration in an ERK-dependent manner. **a** Distinct cellular responses occur in planarians undergoing wound healing (after H-wounds) or regeneration (after R-wounds). **b** "Dormant" tails that were re-amputated, whether immediately or 5 days after drug removal, fully regenerated. **c** Tail fragments did not regenerate after a 5-day PD treatment and recovery period unless re-injured; as in the case of re-amputations, incisions that cut through the animal along the DV axis but do not involve tissue loss led to the formation of the anterior regeneration pole (marked by *notum*; sample numbers indicated in each image) and full regeneration of heads and medial structures (see Supplementary Fig. 6 for representative images of immediate wounds). During the recovery period (blue in scheme), animals were kept in DMSO-containing water. **d** Induction of regeneration through re-injury was inhibited by PD treatment. **e** Animals that had formed blastemas were considered to be "regenerating" if they later also regenerated eyes. **f**–**g** Corresponding proliferative responses at 4 d.p.a. (**f**) and functional recovery (**g**) were observed. Bars in graphs represent mean ± s.d.; two-sided *t* test (compared with uninjured PD-removed animals (first bar, dark gray), unless indicated otherwise), $*p < 0.05$, $**p < 0.01$, $***p < 0.001$, NS, not significant. **h** Increased pERK levels were observed at 3 h post-wounding in conditions that induced regeneration. d.p.w. days post-wounding, scale bars: 200 μm

positional information in the tissue. Interestingly, previous work had shown that planarians depleted of *β-catenin-1* form accessory heads at all R-wound sites[28–30], even when they involve minor tissue loss[30]; H-wounds have not been tested in this context. Hence, we used *β-catenin-1* RNAi planarians as an independent

tissue paradigm to test for the capacity of H-wounds to similarly initiate outgrowths. Remarkably, H-wounds inflicted on intact *β-catenin-1* RNAi animals induced outgrowths in an ERK-dependent manner. These structures had anterior identity and in some cases, developed into heads, including eyes (Fig. 4a, b).

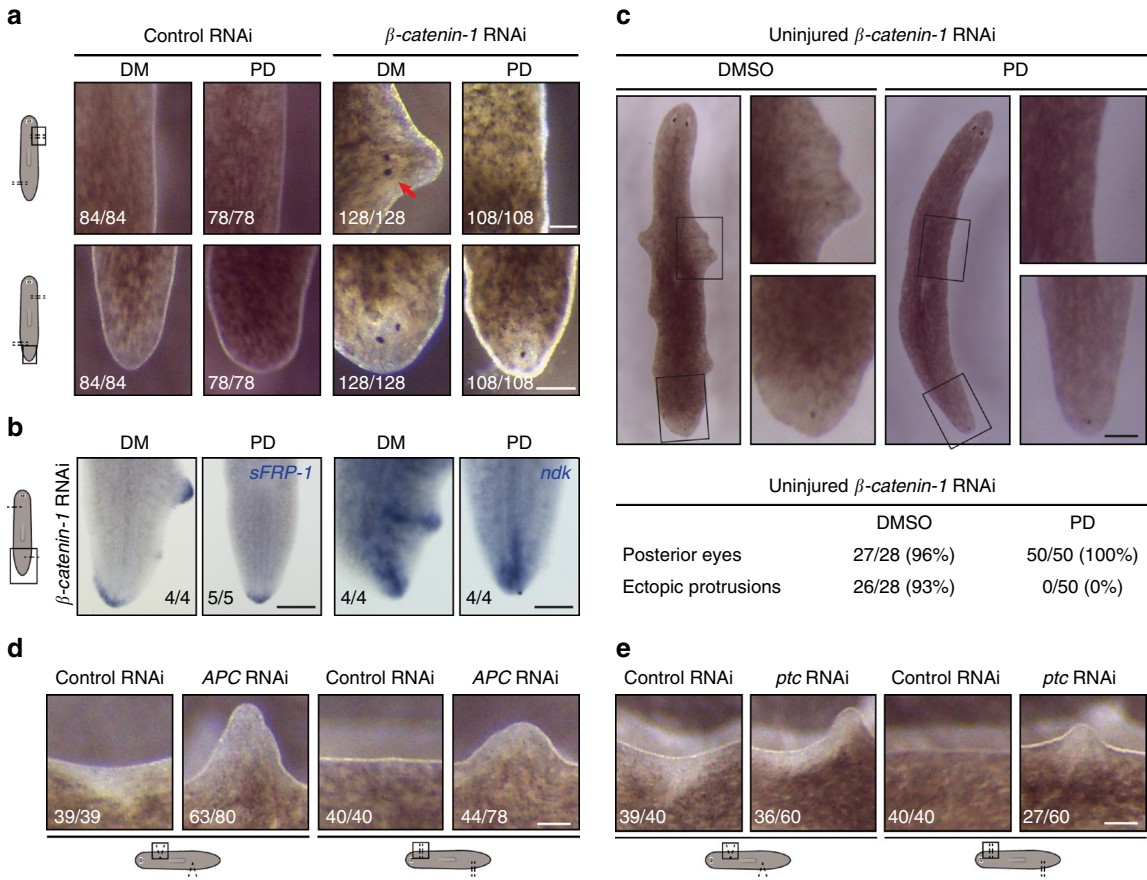

**Fig. 4** H-wounds inflicted on planarians with altered levels of Wnt signaling induce patterned outgrowths. **a, b** Simple incisions applied to *β-catenin-1* RNAi planarians induced accessory heads, which were not observed when ERK activation was inhibited by PD, or in control RNAi animals. Sample numbers indicated in each image. In contrast, ectopic heads formed in existing posterior tissues were unaffected by PD treatment. Arrow, ectopic eyes; *sFRP-1*, anterior marker; *ndk*, marker for the brain region. **c** Similarly, ectopic lateral outgrowths but not posterior heads in homeostatic *β-catenin-1* RNAi animals were inhibited by treatment with PD. **d–e** Outgrowths were also induced by lateral incisions inflicted on *APC* (**d**) or *patched* (**e**) RNAi planarians, which undergo direct[28–30] or indirect[31] ectopic activation of Wnt signaling. To increase the penetrance of outgrowth formation in (**a**), (**d**), and (**e**), two incisions were made instead of one. Numbers refer to injured areas (two per animal); scale bars: 100 µm

Interestingly, lateral outgrowths that are frequently observed in otherwise intact *β-catenin-1* RNAi planarians[28–30] did not form when these animals were treated with the ERK inhibitor (Fig. 4c). Additionally, H-wounds inflicted on planarians depleted of direct (*APC*)[28] or indirect (Hh inhibitor *patched*)[31,32] inhibitors of the Wnt pathway also produced protrusions (Fig. 4d, e). In summary, our findings suggest that disruption of positional cues induces or mimics the presence of missing-tissue signals in planarians and that the combination of aberrant positional cues such as abnormal levels of Wnt activity, and ERK-mediated wound signals distinguish regeneration from simple wound healing in these animals.

**Generic wound signals can initiate regeneration in zebrafish.**
We next asked whether generic wound signals could also trigger regeneration in a missing-tissue context in vertebrates. The zebrafish fin is an excellent model for vertebrate regeneration: within 3 weeks after amputation, each fin ray regenerates all missing tissues, including bones, independently of other rays in the same fin[33]. Fibroblast growth factor (FGF) signaling, a pathway that signals through ERK[34], is required for fin regeneration, in particular for blastema formation and proliferation, but not for formation of a wound epidermis[35,36]. To generate a missing-tissue context similar to the planarian paradigm, we transiently overexpressed a heat-shock (hs) inducible dominant-negative Fgf receptor 1 in *hsp70l*:dnFgfr1-EGFP[pd1] transgenic fish starting prior to fin amputation and continuing until 8 days post-amputation (d.p.a., Fig. 5a)[37]. This allowed for epidermal wound healing (Supplementary Fig. 8a) but blocked regenerative growth in the vast majority of fin rays (Fig. 5a). Importantly, when we relieved fish from heat-shock, expression of the dnFgfr1-GFP transgene subsided to undetectable levels within 5 days (Supplementary Fig. 8b), yet fin rays failed to resume regenerative growth (Fig. 5a). We conclude that transient inhibition of FGF signaling in zebrafish fins results in permanent inhibition of regeneration, thereby creating a stable missing-tissue context (Fig. 5b), similarly to ERK inhibition in planarians.

We then asked whether such dormant fins maintained the competence to regenerate, by re-amputating individual fin rays (Fig. 5c). Importantly, after 5 days of recovery from heat shocks, re-amputation of individual *hsp70l*:dnFgfr1-EGFP fin rays led to initiation of blastema formation within 3 days, and robust ray growth and bone formation within 7 days (Fig. 5d, asterisk). Hence, these dormant fins had retained the capacity to regenerate in response to R-wounds. Next, we applied H-wounds to individual dormant *hsp70l*:dnFgfr1-EGFP rays by removing the wound epidermis located distally to the bony rays without injuring bony tissue, which represents a minor epidermal injury

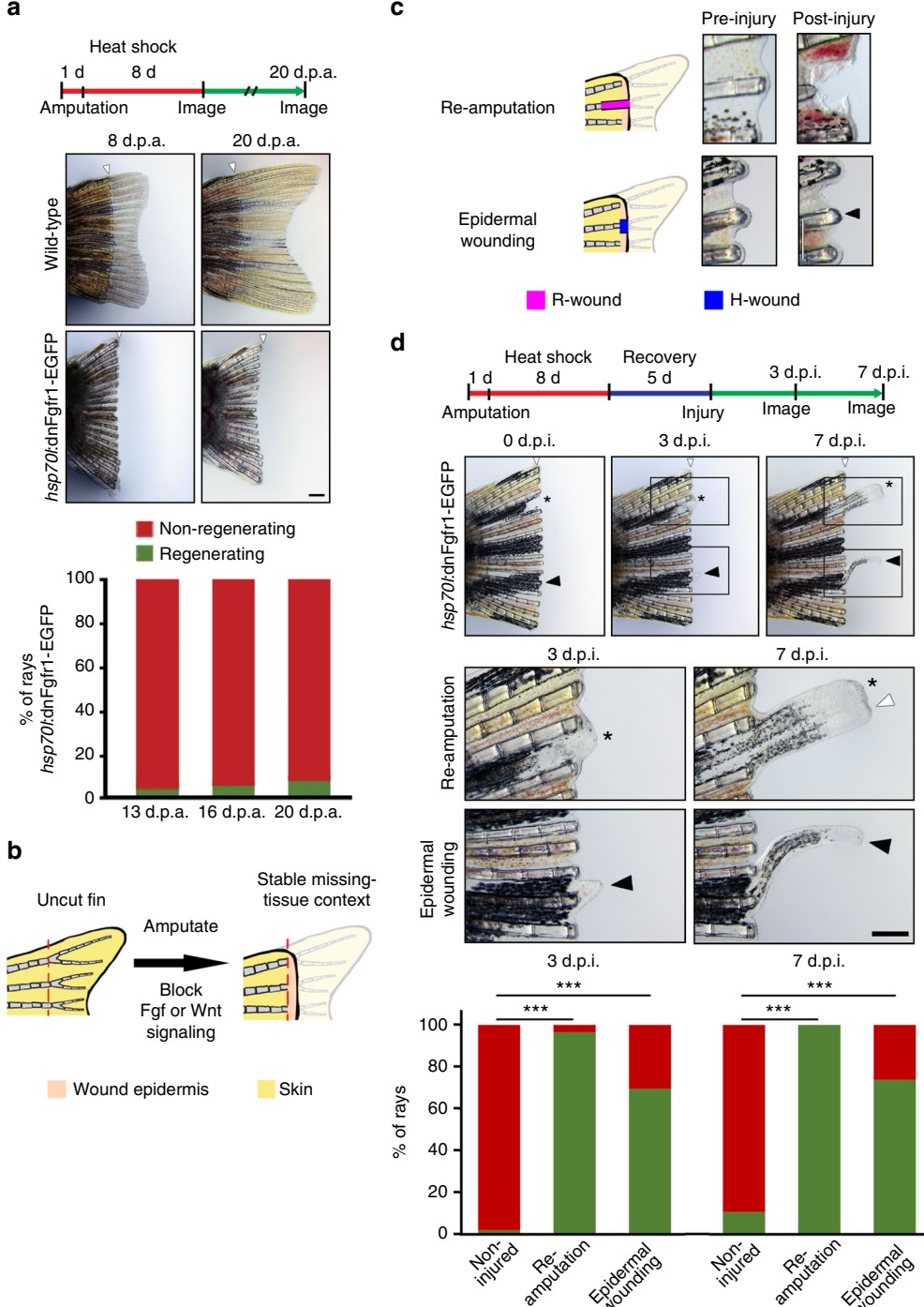

**Fig. 5** Epidermal wounds inflicted on dormant zebrafish fins induce regeneration. **a** Transient overexpression of a dominant-negative Fgf receptor-1 (dnFgfr1-GFP) until 8 days post-amputation (d.p.a.) blocked fin regeneration, which spontaneously resumed only in a minority of fin rays after relief from transgene expression. White arrowheads mark the amputation plane, $n = 9$ fish, 163 rays. **b** A stable missing-tissue context (dormant fin) can be produced by transient blockage of Fgf or Wnt/β−catenin signaling, which allows for wound epidermis formation but stably blocks fin ray regeneration. **c** Injury regimes applied to dormant fins. Re-amputation represents an R-wound, epidermal wounding an H-wound. Arrowhead points at epidermal wound. **d** Re-amputation (asterisk), as well as epidermal wounding (black arrowhead) re-initiated regenerative growth. Lower panels show magnification of boxed areas. The graph depicts the fraction of non-injured, re-amputated and epidermally wounded rays displaying regenerative growth at 3 and 7 days post-injury (d.p.i.). ***$p < 0.001$ $\chi^2$ test. $n$ (non-injured) = 449 rays, 61 fish, 3 experiments; $n$ (re-amputated) = 30 rays, 28 fish, 1 experiment, $n$ (epidermally wounded) = 84 rays, 61 fish, 3 experiments. Note that uninjured rays located next to regenerating rays frequently exhibited bystander growth (white arrowhead in box) in both wounding scenarios (see also Supplementary Fig. 9). Scale bars: 500 μm

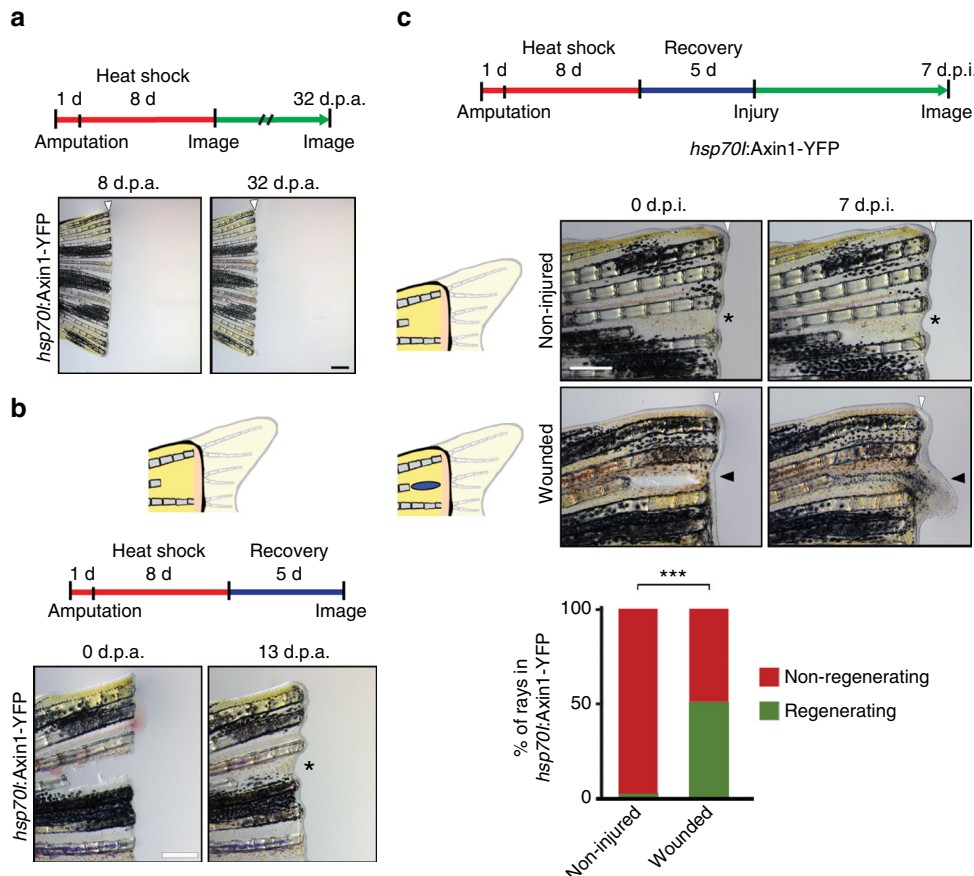

**Fig. 6** The wound epidermis of dormant fins is not a barrier to regeneration. **a** Transient inhibition of Wnt/β-catenin signaling using overexpression of Axin1 in *hsp70l*:Axin1-YFP transgenic fish resulted in a stable blockage of fin ray regeneration. $n = 9$ fish. **b** Amputation and heat-shock regime used to produce recessed rays in dormant fins. Note that interray skin healed distally to non-regenerating recessed rays (asterisk) despite Axin1-YFP expression. **c** Skin wounding applied to the skin distal to recessed dormant rays (black arrowheads) induced regenerative growth beyond the primary amputation plane. Asterisks indicate non-injured control rays. Graph depicts the fraction of regenerating rays. ***$p < 0.001$ $\chi^2$ test. $n$ (non-injured) $= 48$ rays, 48 fish, two experiments, $n$ (wounded) $= 69$ rays, 50 fish, two experiments. Scale bars: 500 μm

(Fig. 5c, arrowhead). When we inflicted similar H-wounds on non-amputated fins, by removing the skin at the distal tip of otherwise intact fin rays, we did not observe outgrowth of bony fin rays, even when rays were monitored for 2 months (Supplementary Fig. 8c). In contrast, when we applied such epidermal H-wounds to dormant fins, 70% of rays developed a blastema by 3 days post-injury (d.p.i.), and 74% displayed regenerative growth by 7 d.p.i. (Fig. 5d, black arrowhead). Regenerative growth induced by these H-wounds was substantially more frequent than spontaneous resumption of regenerative growth in the absence of wounding in non-injured control rays (2% at 3 days and 11% at 7 days; Fig. 5d). Notably, wounding or re-amputation of individual rays induced regeneration of neighboring, non-injured rays at high frequency, but not of the entire fin (white arrowhead in Fig. 5d, Supplementary Fig. 9). Thus, we hypothesize that generic wound signals can spread locally to induce such bystander regeneration, but it appears that they do not act systemically.

We next used a different experimental regime to create dormant zebrafish fin rays. We achieved a permanent block of regeneration when we transiently inhibited the Wnt/β−catenin pathway, which is required for blastema formation, but not wound healing[38,39], using overexpression of Axin1 in *hsp70l*:Axin1-YFP[w35Tg] transgenic fish[40] (Fig. 6a). This time, we created recessed amputations of individual rays and blocked their regeneration using Axin1 overexpression (Fig. 6b). However,

wound healing was not inhibited, resulting in the formation of skin distal to the recessed, non-regenerating rays (asterisks in Fig. 6b, c). Re-amputation of recessed rays resulted in their regeneration in 100% of cases, showing that they had not lost the competence to regenerate (Supplementary Fig. 8d). We then created injuries in the skin distal to the dormant rays, which excluded any chance of inadvertent ray bone injury (Fig. 6c). Remarkably, while only 2% of recessed dormant rays spontaneously resumed regenerative growth in fins that were not re-injured, 52% of rays regenerated upon infliction of skin wounds within 7 d.p.i. (Fig. 6c). Furthermore, regenerating recessed rays were able to grow beyond the primary amputation plane covered by wound epidermis, indicating that the wound epidermis that forms distal to dormant rays is not a physical barrier for ray regeneration (arrowhead in Fig. 6c). Together these results show that, like in planarians, H-wounds are sufficient to initiate regeneration in zebrafish fins when they are inflicted in a missing-tissue context.

## Discussion

The ability to regenerate missing body parts is an awe-inspiring natural phenomenon and a major subject of medical research. While most animals have the ability to heal epidermal wounds, the capacity to regenerate missing body parts is restricted to few animals. The mechanisms governing regeneration initiation and

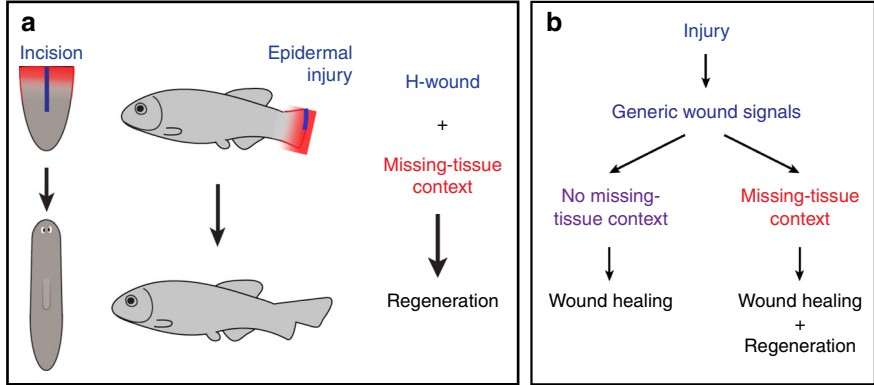

**Fig. 7** Generic wound signals initiate regeneration in missing-tissue contexts. **a** Injuries that normally do not trigger regeneration (H-wounds) do initiate regeneration in planarian dormant tails and zebrafish dormant fins (missing-tissue contexts). **b** Signals that are generically induced after all injuries have the capacity to initiate regeneration, but only when induced in missing-tissue contexts

how they differ from mechanisms initiating wound healing are poorly understood. Many signals that are released soon after epidermal wounding and play important roles in the detection of wounds and other early responses, like the recruitment of immune cells, have been identified and include extracellular ATP, reactive oxygen species (ROS) and polyunsaturated fatty acids (PUFAs)[41]. While some of these signals have also been shown to be activated in response to amputation wounds that trigger regeneration, a very fundamental question has remained unanswered: at which stage after wounding do the responses to wounds that normally initiate healing (H-wounds) differ from those that trigger regeneration of missing body parts (R-wounds)? Do the early cocktails of signals triggered by these wounds already differ (Hypothesis I in Fig. 1)? Or are the early signals shared and do responses diverge due to cues encoded in the tissue? (Hypothesis II in Fig. 1). In this study, we utilized temporally controllable reagents—small-molecule ERK inhibitors in planarians and heat-shock mediated overexpression of transgenes in fish —to generate stable missing-tissue contexts, in which the regeneration-inducing capacity of wound signals could be tested. Our results show that H-wounds are sufficient to trigger regeneration if they are applied in a permissive tissue context, in particular when body parts are missing or positional cues are altered (Fig. 7a). Hence, we conclude that H- and R- wounds trigger the same signals, but depending on the tissue context, these signals activate wound healing only or wound healing followed by regeneration (Fig. 7b).

While our results are consistent with the hypothesis that the initial wound signals triggered by H- and R- wounds are identical (Hypothesis II in Fig. 1), they do not exclude that the amplitude or duration of some early signals triggered by these wounds might differ. In fact, in dormant planarian tails, infliction of multiple H-wounds induced higher levels of phosphorylated ERK than a single R-wound in the same missing-tissue context, but resulted in lower penetrance of rescued animals, suggesting that factors other than pERK levels influence regenerative outcomes. Some evidence for the importance of the duration of wound signals also comes from studies showing that ROS activation, while triggered both by H- and R- wounds in zebrafish fins, lasts for only <2 h in response to H- wounds, but is sustained for >16 h in response to R-wounds[42]. A similar observation has been made in planarians, where some genes are induced at different wound types, yet their expression is only sustained at R-wounds[4]. In the future it will be interesting to identify factors that sense tissue loss and mediate tissue identity and how these can translate identical signals into wound healing or regeneration responses.

Recent studies have uncovered nerve-derived factors and positional cues as essential for the creation of a regeneration-

competent context. In the salamander accessory limb model for instance, the experimental deviation of nerves plus the juxtaposition of different positional cues leads to the formation of accessory limbs[43]. Nerves are required for regeneration also in other systems, for example for zebrafish pectoral fin and heart regeneration[44,45]. The molecular factors mediating the role of nerves have begun to emerge, but interestingly it appears that they differ between systems: while the secreted protein nAG is sufficient to rescue the loss of nerves in salamander limbs, it is unlikely to play a major role as a nerve-derived factor in other species[46,47]. FGF and BMP signals can replace nerve deviation in urodele tail regeneration[48], while nerve growth factor and Neuregulin1 can rescue regeneration of denervated hearts[45]. Thus, while nerve-derived factors are good candidates for programming tissue responses to generic signals toward regeneration, much needs to be learned about their role and to which extent it is conserved between species.

In planarians, inhibition of ERK signaling resulted in animals that were able to identify wounds and activate wound closure, but failed to initiate regeneration. This establishes ERK as essential in the earliest steps of regeneration initiation known in planarians, and suggests that the generic wound response can be divided into an ERK-dependent path toward regeneration and an ERK-independent path leading to wound healing. Notably, ERK-mediated receptor tyrosine kinase (RTK) signaling, such as EGFR and FGFR signaling[49], have been shown to play critical roles in regeneration-related processes, including blastema formation in newt and axolotl limb regeneration[50], transdifferentiation of pigment epithelial cells during retinal regeneration in newts[51,52], as well as dedifferentiation of Müller glia during retina regeneration in zebrafish[19] and of postmitotic salamander muscle cells[20,53]. Thus, targeting ERK signaling cascades may broaden our understanding of how regenerative animals deploy distinct molecular responses depending on whether tissues need to be replaced.

In zebrafish, many signaling pathways have been shown to be required for fin regeneration, in particular for blastema formation and proliferation[11]. While not all of these have been carefully tested for their requirement for epidermal wound healing, the emerging scheme is that the classical pathways regulating embryonic development (e.g., Wnt, FGF, retinoic acid, BMP, Hh, Activin) are dispensable for wound healing, but necessary for regeneration. Here we have used transient blockage of both FGF and Wnt/β-catenin signaling pathways to create dormant fins. However, it is conceivable that transient interference with other morphogenic pathways may similarly prevent regeneration while allowing for wound healing to occur.

In intact planarians depleted of *β-catenin-1*, *APC*, or *ptc*, infliction of lateral H-wounds induced tissue outgrowths, such as

small heads in *β-catenin-1* RNAi planarians. Consistent with the role of ERK signaling in regeneration initiation, these lateral outgrowths were inhibited by PD treatment. In contrast, the differentiation of ectopic heads in the posterior region of *β-catenin-1* RNAi planarians was unaffected by this drug. This suggests that different mechanisms are responsible for inducing the two types of ectopic heads, with ERK being required for the wound-induced outgrowths, but not the wound-independent differentiation process. Although these experiments involve generation of body parts that have never existed, and therefore cannot be defined as regeneration, they are reminiscent of regeneration in that organized outgrowths are formed. Together, these results suggest that disruption of positional cues induces or mimics the presence of missing-tissue signals.

Notably, the observed outgrowths in *β-catenin-1* RNAi planarians are reminiscent of those observed in animals with juxtaposed tissues that originate from different positions along the body axes, such as grafting tail tissues into prepharyngeal areas or reversing the orientation of a graft along the DV axis[54–56]. Such grafts induce 'intercalary regeneration' of tissues that normally lie between the juxtaposed tissues[57]. Importantly, inserting a graft to the position where it was removed from without changing orientation does not result in outgrowths, demonstrating that even after R-wounds, regeneration is only initiated in a permissive tissue context. Together with our findings presented here, these data suggest that first, regeneration initiation signals, regardless of whether they are induced by H- or R-wounds, are only interpreted as regeneration initiation cues in a permissive tissue context, and second, it is molecular cues that indicate tissue identity, rather than the mere physical presence or absence of tissue, that induces the regenerative response.

In summary, we provide a powerful assay—equally suitable for planarians, fish, and likely other regenerating animals—for temporally separating the process of wounding from the loss of body parts. Using this assay, we show that regeneration relies on the conserved principle that generic wound-induced signals have the power to initiate regeneration if they occur in a permissive tissue context (when body parts are missing or when positional identity of tissues is altered) (Fig. 7). Importantly, these signals induce a complete regeneration process that leads to full restoration of missing tissues, including the nervous system and bones. Our study therefore raises the possibility that the wound healing response common to many species could be redirected to regenerative outcomes—by providing the wounded area with proper missing-tissue cues. Whether previously identified molecules with regeneration-inducing capacity[43,58] are among the relevant signals or part of the permissive tissue context will be an important question for future investigations.

## Methods

**Planarian husbandry.** A clonal line of asexual *S. mediterranea* (BCN-10, originally provided by E. Saló) was maintained at 20 °C in Planarian artificial medium (1.6 mM NaCl, 1 mM CaCl$_2$.2H$_2$O, 1 mM MgSO$_4$, 0.1 mM MgCl$_2$, 0.1 mM KCl, and 1.2 mM NaHCO$_3$ in deionised water). The animals were fed 1–2 times per week with calf liver and starved for at least 7 days prior to experiments.

**Small molecule inhibitors and RNAi treatment.** PD0325901 (Calbiochem) and U0126 (Cell Signaling Technology) were dissolved in DMSO, used at 10 and 25 µM, respectively, and replaced daily. Three pulses of 32.2 nl of a solution of 1–1.5 µg/µl dsRNAs were injected ventrally into the intestinal cavity on 6 days over 2 weeks, using a Drummond Scientific Nanoject injector (Broomall, PA, USA), to induce RNAi. They were synthesized through in vitro transcription of PCR products generated using published primers for *activin-1* and *activin-2*[26] or the following primers, shown in 5′–3′ direction:

follistatin: 5′-AGAAATCCACAAGAGGCTGC-3′, 5′-GCATTCAGAAGGCA TTGTCC-3′

APC: 5′-TCTACGGGATCTGCTGCTAC-3′, 5′-CTATCATAGTCATCAGGAT ACG-3′

*β-catenin-1*: 5′-TCCTTCAAATACCAACGCA-3′, 5′-ATCTGGTATCGCACT TTCAGC-3′

patched: 5′-TGTGCGACAATTTTGATGG-3′, 5′-CATCGGTGCCTTTTGT TTT-3′

**Quantitative RT-PCR.** RNA was isolated with TRIzol (Life Technologies) and cDNA was synthesized using the SuperScript III First-Strand Synthesis System (Invitrogen) with Oligo dT primers. Transcript levels were determined using Universal Probe Library (Roche), TaqMan Universal PCR Master Mix (Applied Biosystems), and the Real-Time PCR System 7500 (Applied Biosystems) in three biological replicates. *gapdh* was used as an internal reference gene. The following primers, shown in 5′-3′ direction, and probes were used:

*egr2*: 5′-AGATAGTAATTGTTTCACTGCGTCA-3′, 5′-CATTTTCTTTGAAT CTACATCACTCG-3′ (#2)

*egrl1*: 5′-AATTTAACGAAATCCAAACAGGAT-3′, 5′-TCTGTATTATCGAG ATAGAACGACTCA-3′ (#72)

*gapdh*: 5′-CATCAAAATTGTGGAGAGATGG-3′, 5′-CCAGTTGAAGCCGG AATAAT-3′ (#68)

*mex3*: 5′-TCCAAACTGCACTCAGATTACAA-3′, 5′-GCAGGCAATTCCT CATATCG-3′ (#132)

*notum*: 5′-TGAAGTAGCTTTTGTTGGATCG-3′, 5′-TCTTTTCAGCCTATC GATATTCAA-3′ (#31)

*runt1*: 5′-GCTGATGGCAACGACACTT-3′, 5′-TGATGGGATTGCGAGTTG TA-3′ (#53)

*runt2*: 5′-CGGGAACCCAGATCAAATAA-3′, 5′-GCGCTCAATCAAAATG GAAC-3′ (#145)

*traf2*: 5′-TTGTCAACGCATGCCAGT-3′, 5′-GTTTCCCCTACTCACATAG TCCA-3′ (#5)

*wnt1*: 5′-GCAGTCAAAAGTATGAACGTGAA-3′, 5′-TGGCAAATTTTGGT CGTACA-3′ (#37)

**Antibody generation and western blot analysis.** The rabbit phosphorylated ERK antibody was generated using the peptide GIL(pT)E(pY)VATR[21] and affinity purified (BioGenes). Total protein extracts were prepared by homogenizing snap-frozen animals in RIPA buffer (50 mM Tris-HCl, pH 7.5, 150 mM NaCl, 1% Triton X-100, 0.5% deoxycholate, 0.1% SDS, 1% DTT) containing protease- and phosphatase inhibitors (cOmplete ULTRA and PhosSTOP, Roche) and 1.5 µl Benzonase (Roche) on ice. Extracts were cleared by centrifugation (16,000×*g*, 20 min, 4 °C), concentrated by adding 4 volumes −20 °C acetone and re-dissolved in urea buffer (6 M urea, 2 M thiourea, 20 mM HEPES, pH 7.5)[59] and Bradford reagent (AppliChem), quantified by Bradford reagent (AppliChem), per lane was resolved on a discontinuous 10% SDS-PAGE gel. Following semi-dry electroblotting, the PVDF membrane (Immobilon-P, Millipore) was blocked with 5% bovine serum albumin (BSA, AppliChem) in TBST (20 mM Tris-HCl, pH 7.6, 150 mM NaCl, 0.1% Tween-20) and probed subsequently with the primary (pERK (1:400) and mouse-anti-alpha tubulin (1:90,000; Clone DM1alpha, Neomarkers) or mouse-anti-actin (1:1000, Supernatant, Clone JLA20, DSHB)) and HRP-conjugated secondary (Donkey-anti-mouse or -rabbit, Jackson Immunoresearch, 1:10,000 diluted) antibodies, with intermittent washes in TBST. Target protein bands were visualized using ECL Plus (Amersham) and Hyperfilm ECL (Amersham). All uncropped western blots can be found in the Supplementary Figs. 10–16.

**In situ hybridization on planarian samples.** Planarians were treated with 2% HCl in PBS on ice for 3–5 min, fixed in Carnoy (60% Ethanol, 30% Chloroform, 10% Acetic acid) for 2 h at 4 °C on a shaker and stored in 100% MeOH for at least 2 h at −20 °C before being bleached in 5% H$_2$O$_2$/MeOH for 16–20 h.

Whole-mount in situ hybridizations (WISH) were carried out by hand or using the InsituPro VSi hybridization robot (Intavis). Samples were rehydrated through a series of EtOH/PBS washes (75, 50, 30% EtOH) for 20 min each at 4 °C, washed once with PBST-0.1% (0.1% Triton-X/1X PBS in MilliQ water) for 20 min at 4 °C, and treated with 20 µg/µl Proteinase K (Fermentas) in PBST-0.1% for 8 min at 37 °C. Next, the samples were rinsed three times with ice-cold PBS, post-fixed in 4% PFA/PBS for 1 h at 4 °C, rinsed again with ice-cold PBS and incubated in PBS for 1 h at 4 °C. Samples were then treated with four 15 min incubations: twice with 0.1 M Triethanolamine (TEA) in MilliQ water, followed by 0.25 % acetic anhydride in TEA and 0.5% acetic anhydride in TEA. Following this, samples were rinsed twice with PBSTw-0.1% (0.1% Tween 20/1X PBS in MilliQ water), washed once in PBSTw-0.1% for 10 min and incubated in pre-hybridization buffer (50% Formamide, 5X SSC, 0.1 mg/ml yeast tRNA (Invitrogen), 0.1 mg/ml Heparin, 0.1% Tween 20 and 10 mM DTT in MilliQ water) for 1 h at 56 °C. 110–200 ng of labeled probes in 550 µl of hybridization buffer (10% Dextran sulphate in pre-hybridization buffer) was then heated up to 72 °C for 10 min, cooled down on ice, then incubated with the samples for 16 h at 56 °C. Next, the hybridization buffer was removed, and the samples were rinsed twice with pre-warmed PostHyb (50% Formamide and 5X SSC in MilliQ water), followed by a series of 10 min with dilutions of pre-warmed PostHyb (100–25% PostHyb1 and 0–75% 2X SSC/0.1% Triton), all at 56 °C. This was followed by two 30 min washes with 2X SSC in 0.1% Triton and two 30 min washes with 0.2X SSC in 0.1% Triton, all at 56 °C. Samples were then washed twice for 10 min each with Buffer I (20 mM maleic acid, 30 mM NaCl, 0.1% Triton-X in MilliQ water; pH 7.6), followed by blocking in Buffer II

(1% blocking reagent (Roche) in Buffer I) for 1 h. Next, samples were incubated with an Anti-Digoxigenin-AP, Fab Fragments antibody (1:2000, Roche) in Buffer II for 3 h. After six 10 min Buffer I washes and an overnight incubation in Buffer I, samples were rinsed with TMN (0.1 M Tris-HCl (pH 9.5), 0.1 M NaCl and 1% Tween 20 in MilliQ water). After incubation in 20 μl NBT/BCIP in TMN + 10% PVA in the dark, the reaction was stopped by two PBS washes for 5 min each. Samples were then treated with a 30 min incubation in 4% PFA/PBS, four 5 min washes with PBS, and a series of 10 min EtOH/PBS washes (30, 50, 70, 100, 70, 50, 30% EtOH). Finally, after two 10 min PBSTw-0.1% washes, samples were mounted in Aqua- Poly/Mount. All steps were performed at RT, unless indicated otherwise.

Primers used for in situ probes, shown in 5′-3′ direction, are as follows:

*egr2*: 5′-AAAGAATCCGGCTTCAGACA-3′, 5′-GGTTCGAATTCAGGAGC TGT-3′

*fst*: 5′-AGAAATCCACAAGAGGCTGC-3′, 5′-GCATTCAGAAGGCATTG TCC-3′

*MAT*: 5′-GAATCTGTGGCGAAGGACAT-3′, 5′-AATTGACAAAGGAGC GCAAT-3′

*ndk*: 5′-GTCAAGAGAAGGAACCGACG-3′, 5′-TGACGATGGAGCTTGA GATG-3′

*notum*: 5′-TCGTTGATGATTTGACCCAA-3′, 5′-GAGCGGTGAACATGTT GGATT-3′

*prcn*: 5′-CAAACCCAAAAATTGACCAGCC-3′, 5′-AACCGATATTGGGCT TTTCTGG-3′

*PC2*: 5′-AATCAATGGCATGTCCACCT-3′, 5′-GGGTCGACTCTGTGTTCC AT-3′

*runt1*: 5′-GTAATCCGTCGATGAATAGCG-3′, 5′-CTTTGATGGGATTGCGA GTT-3′

*sFRP-1*: 5′-AAATGACCAAGATTTTTCCCC-3′, 5′-GACCGATTGTTGGGTT TGAC-3′

*wnt1*: 5′-AACAATCAAATATTCGTCA-3′, 5′-TGTTAGGAAAGGTCGG TTGC-3′

**Immunostaining of planarian samples.** Once fixed and bleached as described in the previous section, the animals were rehydrated through a series of MeOH/PBSTx-0.3% washes (75, 50, 25% MeOH in 0.3% Triton-X/1X PBS in MilliQ water) for 10 min each, washed twice with PBST-0.3% for 10 min, and then blocked in 1% BSA in PBST-0.3% for 2 h. Samples were then labeled with rabbit anti-SMEDWI-1 (1:1000), as previously tested in[60], or rabbit anti-phospho-Histone H3 (pH3) (1:600; Millipore) in 1% BSA/PBSTx-0.3% over night at 4 °C. Next, the samples were washed with PBST-0.3% for 1 h six times, then blocked with 1% BSA/PBST-0.3% for 1 h, before being incubated with Alexa Fluor-conjugated secondary antibodies (1:1000; Invitrogen) in 1% BSA/PBST-0.3% over night at 4 °C. This and the following steps were performed in the dark. Samples were then washed with PBST-0.3% for 30 min six times, and incubated in Hoechst/PBST-0.3% for 2 h. Finally, animals were post-fixed in 4% PFA in PBS for 30 min, rinsed with two washes of PBST, and mounted using Aqua-Poly/Mount (Polysciences, Inc.). All steps were performed at RT, unless indicated otherwise. When concanavalin-A[61] labeling was performed with immunostaining, it was added with the secondary antibody. Anti-SMEDWI-1 was generated in rabbit (Biogenes) using a published immunogenic sequence[62].

**TUNEL staining.** Terminal deoxynucleotidyl transferase dUTP nick end labeling (TUNEL) staining was performed as previously described[5]: animals were treated with 10% *n*-acetyl cysteine in PBS for 5 min, fixed in 4% formaldehyde/PBST-0.3% for 20 min and permeabilized 1% SDS/PBS for 20 min. Samples were bleached overnight in 6% $H_2O_2$/PBST-0.3% and rinsed in PBS. The next steps were performed using the TUNEL Apoptosis Detection Kit (Millipore). Samples were incubated with terminal transferase enzyme in reaction buffer for 4 h at 37 °C, then rinsed in stop/wash buffer, followed by 0.25% BSA/PBST-0.3%, and incubated with anti-digoxigenin-rhodamine in blocking solution for 4 h. Finally, animals were rinsed four times in 4 0.25% BSA/PBST-0.3% for 10 min each, incubated in Hoechst/PBST-0.3% for 2 h and mounted using Aqua-Poly/Mount (Polysciences, Inc.). All steps were performed at RT, unless indicated otherwise.

**Image acquisition and analysis.** Planarian live and WISH images were acquired with Leica M80 and Leica M165 FC microscopes, respectively. Z-stacks of fluorescently labeled samples were acquired with a Zeiss LSM700 confocal microscope and processed with the ImageJ package Fiji 1.46r. For quantification of pH3-positive cells, stacks with equal numbers of *z*-sections were taken and cells were counted in the indicated areas using Fiji. Zebrafish bright field and fluorescent images of whole-mount fins were acquired using a Leica M205FA stereomicroscope. Images were processed using Fiji or Adobe Photoshop CS5.1. Contrast/brightness were adjusted exclusively with linear modifications. All modifications were applied equally to the images of control and experimental samples.

**Functional assays for assessing planarian regeneration.** Animals were fed with calf liver. Feeding was considered successful if the food was visible in the gut. For the locomotion assay, a series of images were taken of planarians as they moved over time and the distance traveled measured using Fiji.

**Zebrafish fin injuries and analysis of regeneration.** All procedures involving zebrafish were approved by the Regierungspräsidium Tübingen and Ulm University animal experiment committees.

Transgenic zebrafish and their wild-type siblings of the following lines were used at ~12–18 months of age: *hsp70l*:dnfgfr1a-EGFP[pd1Tg37]; *hsp70l*:Mmu.Axin1-YFP[w35Tg40]. Fin amputations and heat shocks were performed as described[63]. For re-amputation of dormant rays, the distal-most, bony segment was surgically removed. Ablation of epidermis covering individual dormant rays was performed by applying lateral incisions using a scalpel and removing the epidermis with tungsten needles. Incisions in interray skin and in skin distal to recessed dormant rays were applied using a fine tip dissecting knife (Fine Science Tools).

One or two re-injuries (re-amputation or epidermal wounding) were performed per fish, and fish were kept in isolation to avoid re-injury due to aggressive behavior. Fin rays were scored as regenerating at 3 d.p.i. when they contained a blastema that was visible as a condensation of mesenchymal tissue in brightfield images, and at 7 d.p.i. when they contained overt condensed mesenchymal tissue and/or bone. In all re-injury experiments, where fish were heat-shocked from −1 to 8 d.p.a., 4X daily at 37 °C for 1 h, followed by a 5 day recovery period, all fin rays that showed signs of growth at this time point (at 13 days post the initial amputation) were considered escapers which were not completely blocked by dnFgfR1 or Axin1-YFP expression. These rays were excluded from further analysis. To distinguish bystander growth from spontaneously occurring regeneration, only rays that were not located directly adjacent to escapers, to re-amputated, or to epidermally injured rays were considered for the analysis of regeneration of non-injured rays.

For the quantitative analysis of bystander regeneration, rays that grew next to other regenerating rays, and not in isolation, and rays in which the epidermis or the bone was inadvertently injured were excluded. Regeneration of bystander rays was unlikely to be caused by inadvertent undetected injury to neighboring rays, as escapers (uninjured rays that displayed growth at 0 d.p.i.) were associated with bystander regeneration at similar frequency (Supplementary Fig. 9a). A total of 95% of spontaneously regenerating non-injured rays were bystanders, a percentage much higher than expected by chance as determined by Monte Carlo simulation (Supplementary Fig. 9b, c).

**Monte Carlo simulation.** To determine the frequency by which non-injured regenerating rays would be located next to other regenerating rays if spontaneously regenerating rays were randomly distributed, we performed a Monte Carlo simulation[64]. This was based on the number of non-injured rays observed in our wounding experiments 1 and 2, where 142 rays regenerated and 409 did not. We first binary coded ray regeneration as 1 and 0 representing growth and no growth, respectively. To assess the frequency of random occurrence of ray regeneration flanked by no growth (1's flanked by 0's or "010"), we generated a binary random list comprising 142 1's and 409 0's (frequency ~0.258). We then conducted the simulation using 10,000 iterations where each iteration creates a random re-arrangement of 0's and 1's. The frequency of 1's that are flanked by 0's was counted for every round of iteration. P value was obtained by using the formula $P = (r + 1)/(N + 1)$, where $r$ is the number of iterations that had lower "010" frequency compared with the observed empirical frequency in the actual data set and $N$ is the total number of iterations.

**Data availability.** The authors declare that all data supporting the findings of this study are available within the article and its Supplementary Information files or from the corresponding author upon reasonable request.

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

## Acknowledgements

We thank members of the Bartscherer lab for helpful discussions and R. H. Adams, T. Adell, M. Almuedo-Castillo, J. Bruder, F. Cebria, S. Leidel, A. Malapetsas and E. Raz for comments on the manuscript. Research in the Bartscherer lab is funded by the Max Planck Society, the German Research Foundation (BA4044/3-1), and the European Research Council (ERC-2016-StG 716894-IniReg). The Bartscherer lab is part of the Cells-in-Motion Cluster of Excellence (EXC 1003-CiM). The Weidinger lab is supported by the German Research Foundation (CRC 1149 and CRC 1279). A.T., H.R., and S.O. were members of CiM-IMPRS, the joint graduate school of the Cells-in-Motion Cluster

of Excellence (EXC 1003-CiM) and the International Max Planck Research School - Molecular Biomedicine.

## Author contributions

S.O. and K.B. conceived the study; S.O., D.S., and K.B. designed the experiments and wrote the manuscript with comments from all the authors; S.O., D.S., F.R., A.T., and H.R. performed the planarian experiments with the level of commitment reflected in the sequence of authors; F.K. and G.W. designed the zebrafish experiments, F.K. performed these experiments, F.K. and G.W. analyzed zebrafish data and wrote the corresponding text; M.A.M. performed the Monte Carlo simulation; S.M. and L.G. provided methodological advise and were involved in important discussions during the starting phase of the study.

## Additional information

**Competing interests:** The authors declare no competing financial interests.

