## [Peer Review File · Nature Communications]

Reviewers' Comments:

Reviewer #1 (Remarks to the Author)

Manuscript review of Owlarn et al., "Generic wound signals initiate regeneration in missing-tissue contexts"

In this study, Owlarn and colleagues strive to parse out whether regeneration-specific or wound-specific signals are responsible for initiating regeneration in planarian and adult zebrafish fin contexts. They selectively inhibit ERK activity using two pharmacological approaches. Early transient ERK activity inhibition results in a regeneration block, but later re-amputation in these animals induces normal regeneration in an ERK-dependent manner. They find that wounds inflicted on planarians with altered levels of Wnt signaling induce outgrowths even without tissue loss. Finally, they induce wound injuries in zebrafish fins that have an early induced block in regeneration, and find that epidermal wounds inflicted on dormant zebrafish fins are sufficient to induce regeneration.

While these experiments are scientifically sound, several of their main points have been indicated in other contexts or are intuitively understood in the regeneration field. Additionally, no new pathways, mechanisms, or research reagents are described in this study. I like their positing of these findings as a potential system for identifying distinct wound- vs. regeneration-inducing signals, but in my opinion they are not really novel or conclusive. This study seems appropriate for a solid development journal like *Developmental Biology* but not enough new information for a broader interest journal. I've added some comments below intended to be constructive.

Specific comments:

1) p. 6-7. The authors should more specifically address whether ERK signaling is required for actual wound epidermis formation. It is clear that these treatments allow for wound healing, but it is unclear whether their treatments prevent the formation or the establishment of a functional wound epithelium, which would then secrete signals to induce blastema formation. Authors should show wound epidermis markers in untreated and ERK inhibition contexts to show whether a wound epithelium is not established, or whether it is established but incompetent to signal to induce regeneration (for instance, see Chablais and Jazwinska, 2010 as it relates to fins). Also, I am not content with the statement that ERK activation is the most upstream step of regeneration initiation known in planarians. Presumably growth factors are induced to activate pathways leading to ERK activation.

2) It is not a novel concept that a wound injury can induce regeneration in a competent system. For example, that is the basis for accessory limb formation in salamanders. Additionally, it has recently been shown that factors can be sufficient to induce regeneration in missing signal contexts (see Nacu et al., 2016; Makanae et al., 2016). On a similar note, this study's experiments based on (Gurley et al., 2008) and (Rink et al., 2009) are not wholly unexpected and do not contribute much to the story. These animals are already predisposed to form ectopic heads even without the H-wound.

3) The term "Dormant" might be oversold; likely there is a tissue scar of some type or modified/thickened (and less functional) wound epidermis that is not competent for signaling. A reamputation would remove the inactive epidermis and enable formation of a signaling epidermis.

4) p. 10 and Fig. 4. Authors use a heat-shock dom-neg Fgfr1 to block regeneration as reported, but find that regeneration is stably blocked. This uses the same tool and approach but the result is different from the Lee and Poss (2005) result that indicated most rays escape after extended Fgfr

inhibition to regenerate normal length. This might have to do with experimental conditions, but the early report at least indicates the ability to regenerate after a long block. I am not convinced that the nuanced novelty here is noteworthy. On the contrary, from what has been published at this point, it would be truly remarkable if a transient blockade of a signaling pathway were to fully abolish the competence for regeneration of a normally highly regenerative system.

5) In neighboring ray wounding experiments, authors should cite (Murciano et al., 2002). From the results presented in that paper, it is not surprising that a wound injury in a ray may induce a response in a neighboring ray, and it is established that blastemas act as their own mini-regeneration centers and do not communicate laterally across the whole fin (as is mentioned as a potential mechanism in lines 203-207).

Other:

Fig. 2 and S7. It needs to be clearer what the different injuries are. How deep do the H-wounds penetrate and what are the Asterisks?

Line 198 – I am unclear about what “spontaneous regeneration in uninjured rays” means.

Supp. Fig. 9d is unclear. Because the injury is so distal, and that is where normal fin outgrowth occurs, a potential regenerative response may be extremely slow. A more appropriate timepoint to assay may be 2 months instead of 29 dpa.

Authors should more clearly (but briefly) describe or reference the distinctions between PD and UO treatments, and why PD was chosen over UO in most experiments.

Reviewer #2 (Remarks to the Author)

Owlarn et al describe interesting results on the activation of regeneration. Regeneration in planarians was blocked by use of an inhibitor to ERK signaling. These animals failed to induce wound-induced gene expression and to show other hallmarks of a regenerative response. Wash-out of the inhibitor was not enough to cause regeneration to proceed; however, wounding these fragments triggered regeneration. A similar result was reported in zebrafish, in which regeneration was blocked using induced expression of a dominant-negative Fgf-receptor; fin regeneration was blocked, but ray regeneration reactivated by ray amputation or epidermal damaging. The authors conclude that regeneration involves generic wound signaling and a missing tissue context. This builds on prior work and concepts on missing tissue responses and generic wound signaling; however it combines these concepts into a unique and nice result. The manuscript is well written and experiments well presented and controlled.

Specific comments:

-Did ERK inhibition block the initial (6h) mitotic response?

-The picture of UO treatment in Fig 1C is a little unclear-is that debris or a small blastema?

-Figure 1D: did Erk inhibition simply block the elevation of mitoses or in general perturb cell division? i.e., is cell division equal to the level for that region prior to amputation?

-The authors should reference that some generic wound induced genes are required for regeneration (Gavino 2013) and that side incisions of beta-catenin RNAi animals caused head outgrowth (Petersen 08)

-The figure 4 legend refers to Figure S11, which doesn't appear to exist

-The difference between ray amputation and epidermal wounding in Figure 4B and text is not clear, until the helpful Figure 4C is examined. I therefore recommend moving present 4C to prior to present 4B, and ideally adding an image for the individual ray amputation just post-wounding for comparison.

We thank both reviewers for the time and effort taken to review our manuscript and the constructive criticism that helped to improve it. Please find below a point-by-point response to all comments and suggestions.

Reviewer 1

In this study, Owlarn and colleagues strive to parse out whether regeneration-specific or wound-specific signals are responsible for initiating regeneration in planarian and adult zebrafish fin contexts. They selectively inhibit ERK activity using two pharmacological approaches. Early transient ERK activity inhibition results in a regeneration block, but later re-amputation in these animals induces normal regeneration in an ERK-dependent manner. They find that wounds inflicted on planarians with altered levels of Wnt signaling induce outgrowths even without tissue loss. Finally, they induce wound injuries in zebrafish fins that have an early induced block in regeneration, and find that epidermal wounds inflicted on dormant zebrafish fins are sufficient to induce regeneration.

*While these experiments are scientifically sound, several of their main points have been indicated in other contexts or are intuitively understood in the regeneration field. Additionally, no new pathways, mechanisms, or research reagents are described in this study. I like their positing of these findings as a potential system for identifying distinct wound- vs. regeneration-inducing signals, but in my opinion they are not really novel or conclusive. This study seems appropriate for a solid development journal like *Developmental Biology* but not enough new information for a broader interest journal. I've added some comments below intended to be constructive.*

We thank reviewer 1 for their critical assessment of our manuscript and their appreciation of our experimental approach and data quality. We have gratefully taken up the suggestions to improve the manuscript by performing all requested experiments. However, we would like to comment on the reviewer's concern that our findings are not novel enough as they are "intuitively understood in the regeneration field".

The reviewer probably agrees that "intuitively understood" concepts also need to be formally tested before considering them scientific facts. Rather than "intuitively understood", however, our results should be surprising to most people in the regeneration field as even well-established scientists have recently used comparative transcriptomics between H-wounds and R-wounds to "distinguish pathways specific to general injury processes from those required for appendage regeneration" (Monaghan et al, 2012) and search for "[amputation-specific genes], distilled apart from the wound healing gene network" (Knapp et al., 2012). Based on these publications and discussions at regeneration meetings, we would argue that it is rather often intuitively understood that regeneration initiation signals are regeneration-specific, which we have now proven is not the case. Although there has been emerging evidence that in fact some genes induced after H-wounds may be relevant to early regeneration (Wurtzel et al., 2015), to our knowledge, there has never been functional evidence that these include factors that have the capacity to initiate regeneration.

Specific comments:

1) p. 6-7. *The authors should more specifically address whether ERK signaling is required for actual wound epidermis formation. It is clear that these treatments allow for wound healing, but it is unclear whether their treatments prevent the formation or the establishment of a functional wound epithelium, which would then secrete signals to induce blastema formation. Authors should show wound epidermis markers in untreated and ERK inhibition contexts to show whether a wound epithelium is not established, or whether it is established but incompetent to signal to induce regeneration (for instance, see Chablais and Jazwinska, 2010 as it relates to fins).*

We agree with the reviewer that the formation of a functional wound epithelium is one of the first important steps in regeneration initiation and might be affected by inhibition of ERK signalling. Due to a lack of markers for demonstrating the functionality of the wound epithelium in planarians, we addressed this question in the fish. Using antibodies against p63 and β -catenin, we can now show that in fins where FGF signaling is inhibited, a multi-layered wound epidermis forms, but at 3 days post amputation it lacks a clearly distinguishable basal layer, which has formed in wild-type fins by that time-point (new data in Supplementary figure 9a). Since formation of the basal layer occurs at 24 to 48 hours post amputation, this (together with the absence of blastema formation) confirms that early regenerative processes are blocked.

We would like to emphasize that it is irrelevant for the logic of our paper at which time point exactly regeneration is halted in dormant planarian tails or fish fins (pre- or post formation of a fully functional wound epidermis). It only matters that we can create conditions where tissue is missing, but regeneration does not resume spontaneously, which then allows us to test which type of injuries are sufficient to trigger regeneration in the missing tissue context.

Moreover, in response to the reviewer's related comment below, we show in the new Figure 5 that the wound epidermis is not a physical barrier to regeneration.

Also, I am not content with the statement that ERK activation is the most upstream step of regeneration initiation known in planarians. Presumably growth factors are induced to activate pathways leading to ERK activation.

We agree and addressed this by further analysing the nature of the ERK-activating signals. While we did not find convincing evidence that EGFR3, a stem cell resident EGF receptor with a published blastema formation phenotype (Fraguas et al., 2011), acts upstream of wound-induced ERK, we found that wound-induced ERK activation occurs in a stem cell- and protein synthesis-independent manner, indicating that the fast ERK-activating injury cues are not newly made proteins (new data in Figure 1c). Moreover, we can now demonstrate that pERK acts at least in part through *folliculin*-mediated Activin inhibition (new data in Supplementary Figure 4g-h), which is the earliest step currently known to be required for planarian regeneration. We have edited the manuscript to clarify the claim that "ERK activation, by a yet unknown factor, is the most upstream step of regeneration initiation known in planarians" (p7).

2) It is not a novel concept that a wound injury can induce regeneration in a competent system. For example, that is the basis for accessory limb formation in salamanders. Additionally, it has recently been shown that factors can be sufficient to induce regeneration in missing signal contexts (see Nacu et al., 2016; Makanae et al., 2016).

We are aware of the suggested literature, but consider none of the cited papers in conflict with the novelty of our findings.

First, the accessory limb model has typically been used to study the role of nerves in initiating blastema formation, and the role of positional cues in patterning and growing a blastema into a limb (Endo et al., 2015), rather than the nature of the injury required to initiate blastema formation. In particular, it has remained unclear whether solely injuring the epidermis (in contrast to full skin wounding) is sufficient to trigger regeneration in the accessory limb model. Purely epidermal injuries can heal in all organisms, while full skin wounds regenerate in a scar-free manner only in some organisms like salamanders. Thus, full skin wounding in salamanders could trigger signals that are permissive for regeneration, while purely epidermal injuries might not. Therefore, it has remained unclear whether truly generic wound signals (that is signals that should also be present in epidermal wounds in mammals) can trigger regeneration. In our manuscript we now show that epidermal injuries are sufficient to trigger regeneration in dormant zebrafish fins. Thus, our study clarifies this important point and demonstrates that the regeneration-triggering signals are generic.

Notably, (Nacu et al., 2016) showed that presence of posterior tissues/activation of FGF8 and SHH contribute to “sustained [blastema] outgrowth” rather than initiating “blastema formation”, while (Makanae et al., 2016) showed that FGF and BMP signals can replace nerves in urodele tail regeneration. We argue that neither study investigated the nature of the injuries involved in inducing regeneration. Instead, they studied the factors involved in creating a regeneration-competent context. We have now put the two papers into the context of our findings (p10).

Secondly, while the accessory limb model has been quite powerful in showing the importance of nerves and the presumed importance of the juxtaposition of fibroblasts with different positional identity for induction of regenerative responses, it requires several surgical interventions. In contrast, by blocking regeneration, we were able to ask which types of injuries are sufficient to trigger regeneration in an otherwise unperturbed context.

Thirdly, the accessory limb model is only available in salamanders, while the dormant principle should be universal and applicable to many organisms. Thus, we find it of great value to test the ability of generic wounds to trigger regeneration also in other systems.

On a similar note, this study's experiments based on (Gurley et al., 2008) and (Rink et al., 2009) are not wholly unexpected and do not contribute much to the story. These animals are already predisposed to form ectopic heads even without the H-wound.

It is correct that *β-catenin-1* RNAi animals are predisposed to form ectopic heads at all R-wound sites and during homeostatic tissue turnover (Gurley et al., 2008, Peterson and Reddien 2008, Iglesias et al., 2009). However, the effects of H-wounds on these animals have never been reported, to the best of our knowledge (only R-wounds, such as head/tail amputations and wedged cuts). Here, we use the *β-catenin-1* RNAi paradigm to confirm the capability of H-wounds to initiate tissue outgrowth and therefore consider these results an important part of the paper that strengthen our conclusions. We also show that these H-wound-mediated outgrowths (as well as the previously reported spontaneous lateral outgrowths) are ERK dependent (Figure 3a-c), while homeostatic transformations of tail into head tissue are H-wound- and ERK-independent (Figure 3a, lower panel). Moreover, we show that RNAi against *APC* and *patched*, which do not render planarians to form ectopic tissues in the absence of wounds (Rink et al., 2009 and Stückermann et al., 2017), similarly generate tissue outgrowths after H-wounds (Figure 3d-e).

3) The term “Dormant” might be oversold; likely there is a tissue scar of some type or modified/thickened (and less functional) wound epidermis that is not competent for signaling. A reamputation would remove the inactive epidermis and enable formation of a signaling epidermis.

As described above, we do find that the wound epidermis that forms in FGF signaling inhibited fins appears stalled at an early stage of wound response, before it matures into a stage that might be fully functional to support regeneration, but it is not thickened (Supplemental Figure 9a).

In addition, we have now added new data to the manuscript (Figure 5) where we established a new fin injury paradigm that allowed us to test whether tissue scars or a modified wound epidermis represent physical barriers for ray regeneration. We generated recessed dormant rays (individual rays were further proximally amputated than their neighbours). This allowed for wound epidermis formation at the primary amputation plane, while the space distally to the recessed ray was filled in with inter-ray skin tissue. We now show that injuries to this inter-ray skin at a distance from “dormant” rays is sufficient to trigger their regeneration. Thus, any potential tissue scars that would have formed at the amputation plane of the recessed “dormant” ray would not be affected by such injuries. In addition, such regenerating rays can grow beyond the wound epidermis that forms at the primary amputation plane, showing that this is also not a physical barrier.

We use the term “dormant” simply to describe the fact that blocked fin rays or planarian tails still retain the ability to regenerate – it was not meant to imply any particular mechanism or reason for the lack of spontaneous resumption of regeneration in the absence of another trigger.

4) p. 10 and Fig. 4. Authors use a heat-shock dom-neg Fgfr1 to block regeneration as reported, but find that regeneration is stably blocked. This uses the same tool and approach but the result is different from the Lee and Poss (2005) result that indicated most rays escape

after extended Fgfr inhibition to regenerate normal length. This might have to do with experimental conditions, but the early report at least indicates the ability to regenerate after a long block. I am not convinced that the nuanced novelty here is noteworthy. On the contrary, from what has been published at this point, it would be truly remarkable if a transient blockade of a signaling pathway were to fully abolish the competence for regeneration of a normally highly regenerative system.

We would like to emphasize that we actually show that the competence for regeneration is NOT abolished after transient blockage of FGF signaling (or Wnt signaling, which we now also show to have the same effect, see new data in Figure 5). Rather, we show that blocked fins are still competent to regenerate when they experience another injury.

Lee and Poss (Development 2005) show that a certain fraction of fins and rays also do not spontaneously resume regeneration after relieve from dnFGFR1 expression (figure 8 in their paper, only 11 of 24 fins showed regeneration of all rays). They used a regime of a single heat-shock per day and (as we know from personal communication) – a system where water temperature is slowly increased from 28°C to 38°C within one hour. In contrast, we used a significantly different regime of 4 heat-shocks per 24 hours and a system in which temperature is raised from 28° to 37°C within 10 minutes. We have found that the latter (a faster ramp rate) results in stronger activation of the heat-shock promoter and the former (shorter intervals between heat-shocks) will keep transgene expression high over longer time-frames. Thus, it is very likely that FGF signaling is more severely blocked in our regime. We have tested several heat-shock regimes (from 1 to 4 heat-shocks within 24 hours applied for different length of time) and found that less severe heat-shock regimes than the one presented in the manuscript also for us did not result in permanent blockage of regeneration, that is more fins spontaneously resumed regeneration after relieve from heat-shocks. Importantly, we now also show that transient blockage of Wnt/beta-catenin signaling is also sufficient to permanently block (spontaneous) regeneration, without blocking the competence for regeneration in response to another injury trigger (new data in Figure 5).

5) In neighboring ray wounding experiments, authors should cite (Murciano et al., 2002). From the results presented in that paper, it is not surprising that a wound injury in a ray may induce a response in a neighboring ray, and it is established that blastemas act as their own mini-regeneration centers and do not communicate laterally across the whole fin (as is mentioned as a potential mechanism in lines 203-207).

While we agree that Murciano et al 2002 present fascinating data showing that individual rays are individual units of regeneration, we cannot find any data in this paper showing or disproving bystander or long-range effects (whether regeneration of one ray can induce regeneration in neighbouring rays), since as far as we can tell, they only transplanted individual rays with the neighbouring rays left intact. Mari-Beffa et al. (1999) J. Anat 195, 393-405, cut away some rays and left others, so here they could have noticed bystander effects, but they do not mention them nor their absence in that paper. Hence, to our knowledge, bystander effects have not been published in regenerating zebrafish fins.

Other:

Fig. 2 and S7. It needs to be clearer what the different injuries are. How deep do the H-wounds penetrate and what are the Asterisks?

Thank you for pointing this out; we have updated the figure legends to explain the asterisks and clarify that H-wounds cut through the animals along the DV axis. These injuries have been shown to trigger the healing but not the regeneration response (Wenemoser and Reddien, 2010).

Line 198 – I am unclear about what “spontaneous regeneration in uninjured rays” means.

When we relieve fins from dnFGFR1 (or, see the new data in Fig. 5, Axin1) expression, blocked regenerative growth can spontaneously resume without the necessity of a further, additional injury to trigger it. This is what we and Poss and colleagues frequently observe when weaker heat-shock regimes are being used (see point 4 above), and we see a certain low fraction of such spontaneously regenerating rays also with harsher regimes (11% at 7 days in the experiment presented in Figure 4). So while most rays show a permanent block, a few rays do not. This is what we referred to as “spontaneous regeneration in uninjured rays”. “Injury” here refers to the second injury (either re-amputation or wounding), not the initial amputation. Importantly, the rate of regeneration after another injury (either re-amputation, or mere epidermal or skin wounding) is much higher (74% in the experiment in Fig. 4). We have added cartoons to our figures which hopefully aid in clarifying the amputation and injury regimes we have used, and we have re-phrased the text describing these spontaneously regenerating rays on pages 12-13.

Supp. Fig. 9d is unclear. Because the injury is so distal, and that is where normal fin outgrowth occurs, a potential regenerative response may be extremely slow. A more appropriate timepoint to assay may be 2 months instead of 29 dpa.

Thanks for this suggestion. We have repeated the experiment and have monitored fins for 2 months. Again, we could not observe any regenerative growth of fin rays in response to the distal skin injury. See the updated figure in Supplementary Figure 9.

Authors should more clearly (but briefly) describe or reference the distinctions between PD and UO treatments, and why PD was chosen over UO in most experiments.

We initially used two independent inhibitors to establish the role of ERK in early regeneration. We then carried on with PD only as it has been shown to be more selective in other systems (Bain et al., 2007). We now explain the reasons for selecting PD over UO briefly in the text (p.8).

--

Reviewer #2 (Remarks to the Author):

Owlarn et al describe interesting results on the activation of regeneration. Regeneration in planarians was blocked by use of an inhibitor to ERK signaling. These animals failed to induce wound-induced gene expression and to show other hallmarks of a regenerative response. Wash-out of the inhibitor was not enough to cause regeneration to proceed; however, wounding these fragments triggered regeneration. A similar result was reported in zebrafish, in which regeneration was blocked using induced expression of a dominant-negative Fgf-receptor; fin regeneration was blocked, but ray regeneration reactivated by ray amputation or epidermal damaging. The authors conclude that regeneration involves generic wound signaling and a missing tissue context. This builds on prior work and concepts on missing tissue responses and generic wound signaling; however it combines these concepts into a unique and nice result. The manuscript is well written and experiments well presented and controlled.

Specific comments:

-Did ERK inhibition block the initial (6h) mitotic response?

We have now added data that demonstrates that drug treatment does not prevent the induction of the first mitotic response (6h) (Supplementary Figures 4d).

-The picture of U0 treatment in Fig 1C is a little unclear-is that debris or a small blastema?

Thank you for pointing this out; we have replaced the picture with one without debris (Figure 1d).

-Figure 1D: did Erk inhibition simply block the elevation of mitoses or in general perturb cell division? i.e., is cell division equal to the level for that region prior to amputation?

We have now added results that show that proliferation levels in PD- and U0-treated animals at 3 dpa are equal to the level for that region immediately after amputation (Supplementary Figure 4d). In injured animals, proliferation levels are lower after 7d treatment with either drug, but the decrease is only significant after U0 treatment (Supplementary Figure 5f).

-The authors should reference that some generic wound induced genes are required for regeneration (Gavino 2013) and that side incisions of beta-catenin RNAi animals caused head outgrowth (Petersen 08)

Thank you for pointing this out; these citations were erroneously removed in the process of making the manuscript more concise. We have now added the (Gavino et al., 2013) citation and emphasised that (Peterson and Reddien, 2008) have shown that small R-wounds (wedged cuts, which have been shown by (Wenemoser and Reddien, 2010) to cause a regenerative response) along the AP axis induces head outgrowths (Figure 2C, Figure S4M). However, we were unable to find results involving H-wounds.

-The figure 4 legend refers to Figure S11, which doesn't appear to exist

Thank you for pointing this out; we have updated the text accordingly.

-The difference between ray amputation and epidermal wounding in Figure 4B and text is not clear, until the helpful Figure 4C is examined. I therefore recommend moving present 4C to prior to present 4B, and ideally adding an image for the individual ray amputation just post-wounding for comparison.

Thanks for this suggestion. We have added a cartoon that should aid in clarifying the injury regimes and images of re-amputated rays prior to and just post-injury. See the updated Figure 4.

Reviewers' Comments:

Reviewer #1:

Remarks to the Author:

I have read the authors' rebuttal, but my initial opinions remain, and I feel that these opinions would represent those of the larger community of reviewers and readers. There are no new tools or technologies and the language is in my opinion oversold. This study fills in some gaps from other groups' publications with experiments that are scientifically sound but not surprising, novel, or fully conclusive as I had explained. My comments were meant to be constructive and I support its publication elsewhere, just not in a broad interest journal alongside of reports with more conceptual or mechanistic novelty.

Reviewer #2:

Remarks to the Author:

The authors have done a fine job of addressing comments from the prior round of review.

Comments:

Line 123: If the authors have in situ evidence that wound-induced stress response gene were unaffected by Erk inhibition, that would add to the paper

Line 144: should reference Figure 1H rather than 1G

Figure 3a-e: were two incisions required to observe the phenotype? If so, this should be noted with this experiment.

We thank both reviewers for reviewing our manuscript once more and the constructive criticism that helped to improve it. Please find below a point-by-point response to the comments and suggestions.

Reviewer 1

I have read the authors' rebuttal, but my initial opinions remain, and I feel that these opinions would represent those of the larger community of reviewers and readers. There are no new tools or technologies and the language is in my opinion oversold. This study fills in some gaps from other groups' publications with experiments that are scientifically sound but not surprising, novel, or fully conclusive as I had explained. My comments were meant to be constructive and I support its publication elsewhere, just not in a broad interest journal alongside of reports with more conceptual or mechanistic novelty.

We are very grateful for the reviewer's constructive criticism that helped us to improve the paper.

--

Reviewer 2

The authors have done a fine job of addressing comments from the prior round of review.

Comments:

*Line 123: If the authors have *in situ* evidence that wound-induced stress response gene were unaffected by Erk inhibition, that would add to the paper*

We have now added *in situ* expression data on *egr11* expression in DMSO and MEK inhibited planarians after both H- and R-wounds in SFig 6e-f. These data confirm that, while *egr11* is more affected than other wound-induced genes by MEK inhibition, it is strongly wound-induced even in the absence of pERK.

Line 144: should reference Figure 1H rather than 1G

Thank you, we have now corrected this reference.

Figure 3a-e: were two incisions required to observe the phenotype? If so, this should be noted with this experiment.

Thank you, we have added "To increase the penetrance of outgrowth formation in a, d and e, two incisions were made instead of one." to the corresponding figure legend.